# Molecular Characterization of Octopamine/Tyramine Receptor Gene of Amitraz-Resistant *Rhipicephalus* (*Boophilus*) *decoloratus* Ticks from Uganda

**DOI:** 10.3390/microorganisms10122384

**Published:** 2022-11-30

**Authors:** Patrick Vudriko, Rika Umemiya-Shirafuji, Dickson Stuart Tayebwa, Joseph Byaruhanga, Benedicto Byamukama, Maria Tumwebaze, Xuenan Xuan, Hiroshi Suzuki

**Affiliations:** 1Research Center for Tropical Diseases and Vector Control, Department of Veterinary Pharmacy, Clinics and Comparative Medicine, School of Veterinary Medicine and Animal Resources, College of Veterinary Medicine, Animal Resources and Biosecurity, Makerere University, Kampala P.O. Box 7062, Uganda; 2National Research Center for Protozoan Diseases, Obihiro University of Agriculture and Veterinary Medicine, Inada-Cho, Obihiro 080-8555, Japan; 3United Graduate School of Veterinary Sciences, Gifu University, 1-1 Yanagido, Gifu 501-1193, Japan

**Keywords:** acaricide resistance, amitraz, *Rhipicephalus* (*Boophilus*) *decoloratus*, octopamine/tyramine receptor gene, ticks

## Abstract

We previously reported the emergence of amitraz-resistant *Rhipicephalus* (*Boophilus*) *decoloratus* ticks in the western region of Uganda. This study characterized the octopamine/tyramine receptor gene (OCT/Tyr) of amitraz-resistant and -susceptible *R.* (*B*.) *decoloratus* ticks from four regions of Uganda. The OCT/Tyr gene was amplified from genomic DNA of 17 *R.* (*B*.) *decoloratus* larval populations of known susceptibility to amitraz. The amplicons were purified, cloned and sequenced to determine mutations in the partial coding region of the OCT/Tyr gene. The amplified *R.* (*B*.) *decoloratus* OCT/Tyr gene was 91–100% identical to the *R.* (*B*.) *microplus* OCT/Tyr gene. Up to 24 single nucleotide polymorphisms (SNPs) were found in the OCT/Tyr gene from ticks obtained from high acaricide pressure areas, compared to 8 from the low acaricide pressure areas. A total of eight amino acid mutations were recorded in the partial OCT/Tyr gene from ticks from the western region, and four of them were associated with amitraz-resistant tick populations. The amino acid mutations M1G, L16F, D41G and V72A were associated with phenotypic resistance to amitraz with no specific pattern. Phylogenetic analysis revealed that the OCT/Tyr gene sequence from this study clustered into two distinct groups that separated the genotype from high acaricide pressure areas from the susceptible populations. In conclusion, this study is the first to characterize the *R.* (*B.*) *decoloratus* OCT/Tyr receptor gene and reports four novel amino acid mutations associated with phenotypic amitraz resistance in Uganda. However, lack of mutations in the ORF of the OCT/Tyr gene fragment for some of the amitraz-resistant *R.* (*B*.) *decoloratus* ticks could suggest that other mechanisms of resistance may be responsible for amitraz resistance, hence the need for further investigation.

## 1. Introduction

*Rhipicephalus* (*B*.) *decoloratus* is a one-host tick that is widely distributed in Africa [1,2,3]. This tick is an important vector for *Babesia* spp. and *Anaplasma* spp., which causes bovine babesioisis and anaplasmosis, respectively [4,5]. The burden and importance of the *R.* (*B*.) *decoloratus* tick have been scantly reported in Uganda [6,7,8]. Chemicals (acaricides) are the common means for tick control and prevention of the diseases vectored by ticks. The emergence of resistance against synthetic pyrethroids (SP) and its co-formulations with organophosphates (OP) makes amitraz a cornerstone for tick control in Uganda. Amitraz is one of the most widely used acaricides for tick control in Uganda [9,10] and in the rest of Africa [11,12,13]. Amitraz belongs to a group of pesticides known as formamidines [14,15]. Formamidines achieve their pharmacological activity by modulating tick octopamine receptors [16,17]. The octopamine receptor is a member of the G-protein coupled receptor (GPCR) that is analogous to the adrenergic receptors in vertebrates [18]. The arthropod octopamine receptors are classified into three, namely alpha (α) and beta (β) adrenergic-like and octopamine/tyramine receptors [19,20,21]. Amitraz causes impairment of tick nervous functions of octopamine and it inhibits oviposition [22]. This unique mode of action of amitraz makes it key in the acaricide rotation against ticks that are resistant to SP and OP acaricides. While there are few reports on amitraz resistance in *R*. (*B*.) *decoloratus*, on the contrary, amitraz resistance is widely reported in *R.* (*B*.) *microplus* from Australia, South America, India and South Africa [12,23,24,25]. The genetic basis of amitraz resistance in *R.* (*B*.) *microplus* was first attributed to mutations in the octopamine receptor [26]. This was confirmed by researchers in South Africa who found that mutations in the octopamine/tyramine (OCT/Tyr) gene, confered resistance against amitraz in South African *R.* (*B*.) *microplus* tick strains [12]. The same mutation in *R.* (*B*.) *microplus* OCT/Tyr that confered amitraz resistance was also associated with amitraz resistance in *R*. (*B*.) *decoloratus* ticks from South Africa [12].

In Uganda, we previously reported the emergence of amitraz-resistant *R*. (*B*.) *decoloratus* ticks in western cattle corridor [27]. However, there is lack of data on a molecular basis of amitraz resistance in Ugandan strains of *R*. (*B*.) *decoloratus* ticks. Genetic techniques have been widely used to investigate the mechanisms of resistance in arthropod vectors [28,29,30]. In the present study, we characterized the *R*. (*B*.) *decoloratus* OCT/Tyr receptor gene from both amitraz-susceptible and -resistant larval tick populations and identified novel mutations that are associated with phenotypic resistance against amitraz in tick populations from Uganda.

## 2. Materials and Methods

### 2.1. Tick Population

The ticks used in this study were partly retrieved from archived samples previously collected from cattle farms in Uganda [27,31]. A total of 17 *R*. (*B*.) *decoloratus* larval tick populations were used. The ticks were categorized into two, based on the region and intensity of acaricide use. The first group were those from farms in central (2) and western Uganda (13) referred to as ticks from high acaricide pressure area. The second category was an amitraz-susceptible population collected from eastern (01) and northern (01) Uganda in an area of low acaricide pressure. The districts from which tick samples were collected is shown in Figure 1.

The farms in the high acaricide pressure region mainly kept crosses of exotic dairy cattle that are susceptible to tick-borne diseases, while those from the low acaricide pressure region kept indigenous cattle breed (Zebu cattle). The engorged female ticks collected from the above farms were reared at the Research Center for Tropical Diseases and Vector Control (RTC) Laboratory, College of Veterinary Medicine, Animal Resources and Biosecurity (COVAB), Makerere University. The susceptibility of the larvae against amitraz was determined by the Larval Packet Test (LPT) as previously reported [27]. The characteristics of the farms and tick populations investigated is shown in Table 1.

### 2.2. Extraction of Genomic DNA

For each tick population, genomic DNA was extracted from approximately 30 pooled larvae from using a Nucleospin Tissue DNA extraction kit (Marcherey-Nagel, Duren, Germany). The larvae that were preserved in 75% ethanol were washed with 1 × PBS and crushed in a Bio-masher II tube (Nippi, Japan) with Bio-masher motor (Nippi, Japan). The DNA was extracted using the above kit following the manufacturer’s instructions. The concentration of the extracted DNA was determined with a Nano Drop 2000 (Thermo Fisher Scientific Inc, Waltham, MA, USA) and a working stock of 40 ng/µL was prepared and stored at −30 °C until use. All the DNA samples used had a purity ratio of above 1.8. The control DNA from *R.* (*B*.) *microplus* was extracted from adult ticks collected from north-eastern Thailand archived at the National Research Center for Protozoan Diseases (NRCPD), Obihiro University of Agriculture and Veterinary Medicine-Japan.

### 2.3. Amplification of Octopamine/Tyramine Receptor Gene

The octopamine/tyramine receptor gene was amplified using a Blend Taq^®^ plus polymerase kit (Toyobo, Japan). The primers designed by previous researchers were used to amplify the region of the OCT/Tyr gene that is susceptible to mutation [26]. These primers were OAR-F171: 5’-GGTTCACCCAACCTCATCTCTGAA-3’ (forward) and OAR-R587: 5’-GCAGATGACCAGCACGTTACCG-3’ (reverse). Amplification was carried out in a 40 µL reaction volume containing 0.2 mM dNTP, 0.24 mM of each primer and 2 ng of genomic DNA template. The thermal cycling conditions included 1 min of initial denaturation at 94 °C and 40 cycles of 94 °C for 5 s and 55 °C for 1 min and further extension of 68 °C for 5 min. The resultant PCR products were electrophoresed in 1.5% agarose gel, stained with ethidium bromide and visualized under UV lamp.

### 2.4. Cloning and Sequencing of Octopamine/Tyramine Receptor Gene

The OCT/Tyr PCR amplicons were purified with Wizard^®^ SV Gel and a PCR Clean-up System (Promega, Madison, WI, USA) according to the manufacturer’s instruction. The gene was ligated in pGEM-Teasy (Promega, Madison, WI, USA) at 4 °C overnight and transformed into ECOS^TM^ competent *Escherichia coli* DH5α (Nipon gene, Tokyo, Japan). For each sample, successful clones were confirmed by colony PCR and at least 4 colonies were multiplied in Luria-Bertaini broth (with 100 µg/mL ampicillin) and purified using a NucleoSpin^®^ Plasmid Easy Pure kit (Marcherey-Nagel, Duren, Germany) following the manufacturer’s instruction. The OCT/Tyr gene insert was sequenced with T7 promoter (forward) primer using a BigDye v3.1 Terminator Cycle Sequencing Kit and the 3730 × l DNA Analyzer (Applied Biosystem, Waltham, MA, USA).

### 2.5. Sequence Analysis

The nucleotide sequence was edited with DNA star (Version 7.1.0) and the resultant sequence was analyzed with BLAST (http://blast.ncbi.nlm.nih.gov/Blast.cgi, (last accessed on 30 November 2022) to determine percentage identity. Mutations in the coding region of the OCT/Tyr gene was analyzed by multiple sequence alignment of the corresponding nucleotide and amino acid sequences using GeneDoc ver 2.7.000. Nucleotide diversity in both acaricide-susceptible and -resistant ticks was compared and their evolutionary history was determined using MEGA5.0 software [32]. The resultant *R*. (*B*.) *decoloratus* Oct/tyramine nucleotide sequences generated in this study were deposited in GenBank under the Accession # ON792209- ON792255.

### 2.6. Ethical Consideration

This study was approved by College of Veterinary Medicine, Animal Resources and Biosecurity, Makerere University (No. VAB/REC/15/104). The DNA experiments were carried out according to ethical guidelines for the use of DNA samples permitted by the Obihiro University of Agriculture and Veterinary Medicine under approval numbers 1217-2 and 1218-2.

## 3. Results

### 3.1. Amplification and Sequence of OCT/Tyr Receptor Gene

The amplified *R.* (*B*.) *decoloratus* partial octopamine/tyramine (OCT/Tyr) gene was 418 bp. The size of the OCT/Tyr PCR amplicon for *R.* (*B*.) *decoloratus* was comparable with that of the *R.* (*B*.) *microplus* OCT/Tyr receptor gene (Figure 2).

BLAST analysis of the sequenced OCT/Tyr open reading frame (ORF) (231 bp) revealed that the OCT/Tyr genes from *R.* (*B*.) *decoloratus* from western and central Uganda were 91–93% identical to the *R.* (*B*.) *microplus* OCT/Tyr gene sequence in GenBank (Accession #: KR081360.1). However, those from the north and eastern region were 95–100% identical to the *R.* (*B*.) *decoloratus* OCT/Tyr nucleotide sequence. Interesting, the *R.* (*B*.) *decoloratus* OCT/Tyr gene nucleotide sequence was 91–97% identical to the *R.* (*B*.) *microplus* GPCR sequence deposited in the GenBank Accession #: AJ010743.1 (Figure 3).

### 3.2. Analysis of SNPs in the Coding Region of OCT/Tyr Receptor Gene

The *R.* (*B*.) *decoloratus* OCT/Tyr partial ORF revealed variable levels of nucleotide similarity ranging from 93.1–100.0% among the different clones and tick populations. Overall, 32 single nucleotide polymorphism (SNPs) were recorded amongst *R.* (*B*.) *decoloratus* OCT/Tyr gene sequences in this study. However, 24 (75%) of the above mutations were only found in ticks from high acaricide pressure areas (HAPA) of western and central Uganda. In contrast, only 8 (25%) SNPs were found in the OCT/Try receptor gene from the *R.* (*B*.) *decoloratus* ticks collected from low acaricide pressure areas (LAPA) in north and eastern Uganda. Only two SNPs were shared between OCT/Tyr genes from ticks collected from Serere (LAPA) and HAPA. Eleven of the above SNPs resulted in a change of amino acid (non-synonymous) in the OCT/Tyr receptor gene (Table 2).

Novel non-synonymous SNPs were observed at four different loci in the ORF for each of the four *R.* (*B*.) *decoloratus* resistant tick populations; A1G (KMGR), C46T (2MTM), A122G (1BUS) and T215C (3MTM) (Figure 3). Based on previous nomenclature, A1G, C46T, A122G and T215C corresponds to A135G, C181T, A257G and T350C, respectively [12,26]. Two additional identical non-synonymous SNPs, C215T and C230T were also found in the *R.* (*B*.) *decoloratus* OCT/Tyr gene from the HAPA with no particular pattern. However, all the non-synonymous SNPs in OCT/Tyr from the amitraz-susceptible *R.* (*B*.) *decoloratus* ticks were not related to those from the high acaricide pressure area as shown in Table 2 above.

### 3.3. OCT/Tyr Gene Amino Acid Mutations in Amitraz-Resistant Ticks

Overall, the *R.* (*B*.) *decoloratus* OCT/Tyr receptor amino acid sequence for the partial coding region were 90.9–100.0% identical. A total of eight amino acid substitutions (M1V, L16F, V27A, V32A, M39L, D41G, I67V, and V72A) were recorded in the coding region of the *R.* (*B*.) *decoloratus* OCT/Tyr gene from the high acaricide pressure area (Figure 4).

Four of the amino acids (M1V, L16F, D41G, V72A) out of the eight mutations corresponded to the novel non-synonymous SNPs found amongst amitraz-resistant tick populations from western Uganda. The amino acid mutation at position 16 (leucine to phenylalanine) was associated with 58.5% survival amongst pooled larvae from farm 2MTM as previously reported by our research team [27]. Low survival rate of 31.9% was associated with mutation at position 41, which led to a change from aspartate to glycine (D41G) in ticks from farm 1BUS. However, amino acid substitution at position 72 (valine to alanine) was associated with 55% survival of larvae against amitraz in tick population from farm 3MTM. Two identical amino acid substitutions at position 27 and 32 (valine to alanine) that corresponds to the C215T and C230T SNPs were recorded in the *R.* (*B*.) *decoloratus* OCT/Tyr gene from the high acaricide pressure area. Interestingly none of the amino acid mutations in the *R.* (*B*.) *decoloratus* OCT/Tyr gene from the susceptible tick populations were related to those from the high acaricide pressure area (Table 2).

### 3.4. Evolutionary Characteristics of R. (B.) decoloratus OCT/Tyr Gene from Resistant Ticks

Phylogenetic analysis revealed that the partial *R.* (*B*.) *decoloratus* OCT/Tyr gene clustered into two distinct groups (Figure 5). Group I consisted exclusively of the OCT/Tyr gene from ticks from the high acaricide pressure area. However, group II was formed by the OCT/Tyr gene from the amitraz-susceptible tick population and the susceptible NCBI reference sequence. Similarly, the *R.* (*B*.) *microplus* GPCR also clustered in the same group with the susceptible *R.* (*B*.) *decoloratus* OCT/Tyr gene for susceptible tick population from eastern and northern region of Uganda.

## 4. Discussion

We previously reported the emergence of the amitraz-resistant tick population in the greater Bushenyi area in Uganda through phenotypic assays. Amitraz is considered as a cornerstone in the control of resistant ticks because it is the choice of acaricide for control of orgnophosphate- and pyrethroid-resistant ticks given its unique mode of action. The OCT/Tyr homologue reported in this study is one of the target sites for amitraz according to previous studies [12,19]. The current study found greater homology between the *R.* (*B*.) *decoloratus* OCT/Tyr receptor gene and that of *R.* (*B*.) *microplus*. This suggests that the gene is highly conserved amongst the two sub-species of Boophilus ticks. Since there was no *R.* (*B*.) *decoloratus* OCT/Tyr sequence in the GenBank, the homology of the current sequence with other *R.* (*B*.) *decoloratus* OCT/Tyr could not be determined. However, we found contrasting number of SNP loci in the *R.* (*B*.) *decoloratus* OCT/Tyr gene from amitraz-resistant and -susceptible tick populations.

The high number of non-synonymous SNPs in the *R.* (*B*.) *decoloratus* OCT/Tyr gene from the western region is suggestive of selection pressure caused by frequent exposure of ticks to amitraz. Selection pressure increases the rate of resistance alleles in ticks [23]. Up to eight amino acid substitutions were observed in the OCT/Tyr gene from amitraz-resistant ticks from western Uganda. However, four of the above substitutions, M1V, L16F, D41G and V72A, were strictly found in the *R.* (*B*.) *decoloratus* tick populations that were resistant to amitraz. Overall, all the amino acid mutations observed in this study were unique from those previously reported in amitraz-resistant *R.* (*B*.) *microplus.* Chen and his colleagues first reported the T8P and L22S as resistance SNPs, which was later confirmed by other researchers in South Africa, India and Brazil [12,26,33,34]. However, this contrasts with our findings in which each of the resistant tick populations had more than one SNP loci that were not related to previous amino acid substitutions reported in South Africa [12]. What remains unclear is whether the various non-synonymous SNPs identified in amitraz-resistant ticks play a key role in amitraz resistance in *R.* (*B*.) *decoloratus* ticks from Uganda.

The L16F mutation was associated with 58.5% amitraz survival by LPT and history of consistent use of amitraz once a week for 2 years. However, D41G and V72A mutation was associated with 31.9% amitraz survival and intermittent use of amitraz. Interestingly, the V72A mutation was associated with 55% amitraz survival rate, yet the history of acaricide use showed that the farmer did not use amitraz for the last 2 years (Table 1). This could be attributed to new animals brought on the farm with ticks that were resistant to amitraz, or other factors that could introduce resistant ticks into the farm. We previously reported that cattle trade and the lapse in inspection and quarantine of purchased animals is amongst the risk factors for spreading acaricide-resistant ticks in Uganda [10,27].

The current study also found two identical amino acid substitutions (V to A) at two loci, 27 and 32, in the OCT/Tyr genes of *R.* (*B*.) *decoloratus* ticks from western Uganda. There were no particular patterns in the occurrence of the above mutations between resistant and susceptible genotypes from the high acaricide pressure area. Surprisingly, one of the tick populations (KKS) that showed 100% survival at a discriminating dose of amitraz did not have unique amino acid substitutions in the OCT/Tyr receptor. This further provides hints that amitraz resistance in *Boophilus* ticks may be mediated via multiple pathways, with OCT/Tyr target site mutation being possibly one of the several mechanisms. Several researchers have highlighted the potential role of other G-protein coupled receptors such as beta- and alpha-adrenergic-like-octopamine receptors and various metabolic enzymes in amitraz-resistant *Rhipicephalus* ticks [12,33,35]. Indeed researchers from South Africa previously identified several transcripts of enzymes from amitraz-resistant ticks that were upregulated following exposure to amitraz [36]. This implores the need for further investigations to understand the mechanism of amitraz resistance using both functional genomics and metabolomics.

In the current study, phylogenetic analysis revealed that the Ugandan strain of *R. (B.) decoloratus* OCT/Tyr genes clustered into two distinct groups that separated the genes from high acaricide pressure areas (central and west) from the susceptible populations (east and north). Of concern was the interspersal of resistant genotypes within the susceptible ones in the high acaricide pressure areas. This could imply that the non-synonymous SNPs observed in the OCT/Tyr receptor coding region is an early indicator for potential amitraz resistance development. It should be noted that there has been prolonged irrational use of various acaricides, including amitraz in Uganda. Therefore, to gain more understanding about the role OCT/Tyr receptor in amitraz resistance in *R. (B.) decoloratus* ticks, we recommend further research using a large number of amitraz-resistant tick populations with various levels of resistance. However, at the farm level, we recommend farmer training on proper acaricide application and rotation across the country. This will help to preserve the efficacy of amitraz, given that resistance to OP and its co-formulation with synthetic pyrethroids is wide spread in Uganda [27].

In conclusion, this study characterized the *R.* (*B*.) *decoloratus* OCT/Tyr receptor gene from both amitraz-susceptible and -resistant Ugandan tick strains. For the first time, we report four unique amino acid substitutions in the *R.* (*B*.) *decoloratus* OCT/Tyr receptor gene that are associated with phenotypic resistance against amitraz, although we postulate that other mechanisms of resistance may play a significant role in amitraz resistance. The finding from this study is expected to stimulate further research on the molecular and biochemical basis of amitraz resistance in *R.* (*B*.) *decoloratus* ticks in Africa and beyond.

## Figures and Tables

**Figure 1 microorganisms-10-02384-f001:**
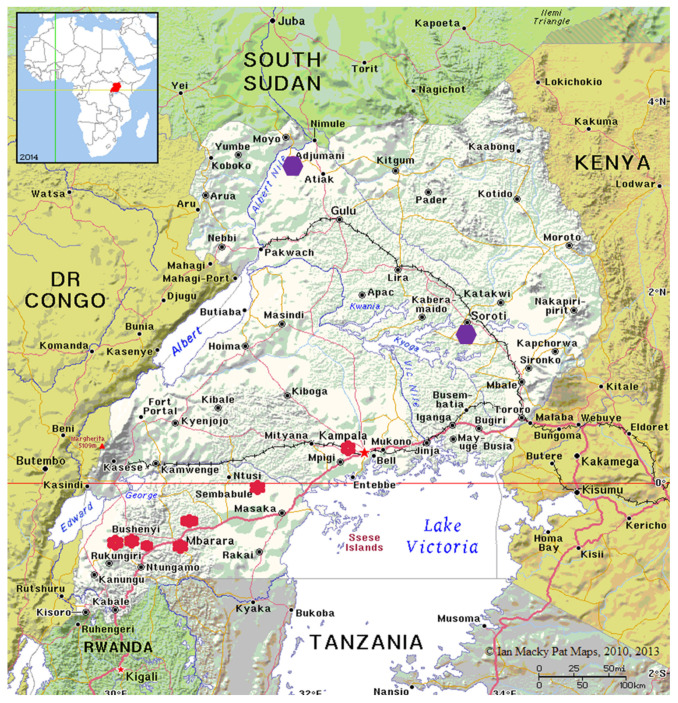
Map of Uganda showing the districts from which tick samples were collected. The red spots represent high acaricide presure area and the purple spot is for low acaricide presure area where the susceptible population was collected.

**Figure 2 microorganisms-10-02384-f002:**
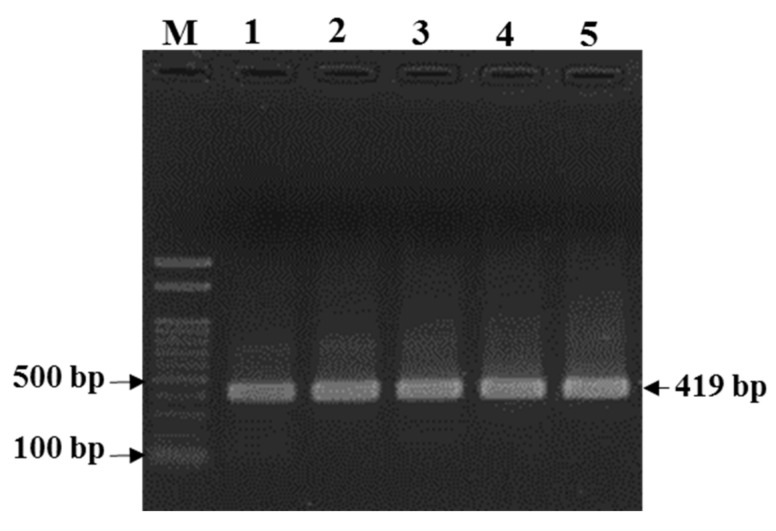
PCR amplification of *Rhipicephalus* (*Boophilus*) *decoloratus* partial octopamine/tyramine receptor gene. Lane M, 100 bp marker; Lane 1, *R.* (*B*.) *microplus* Thailand strain (control DNA); Lanes 2–5, *R.* (*B*.) *decoloratus* DNA samples from different farms/populations from Uganda: 2, 1SHM; 3, 2BUS; 4, WKB; 5, 2MTM.

**Figure 3 microorganisms-10-02384-f003:**
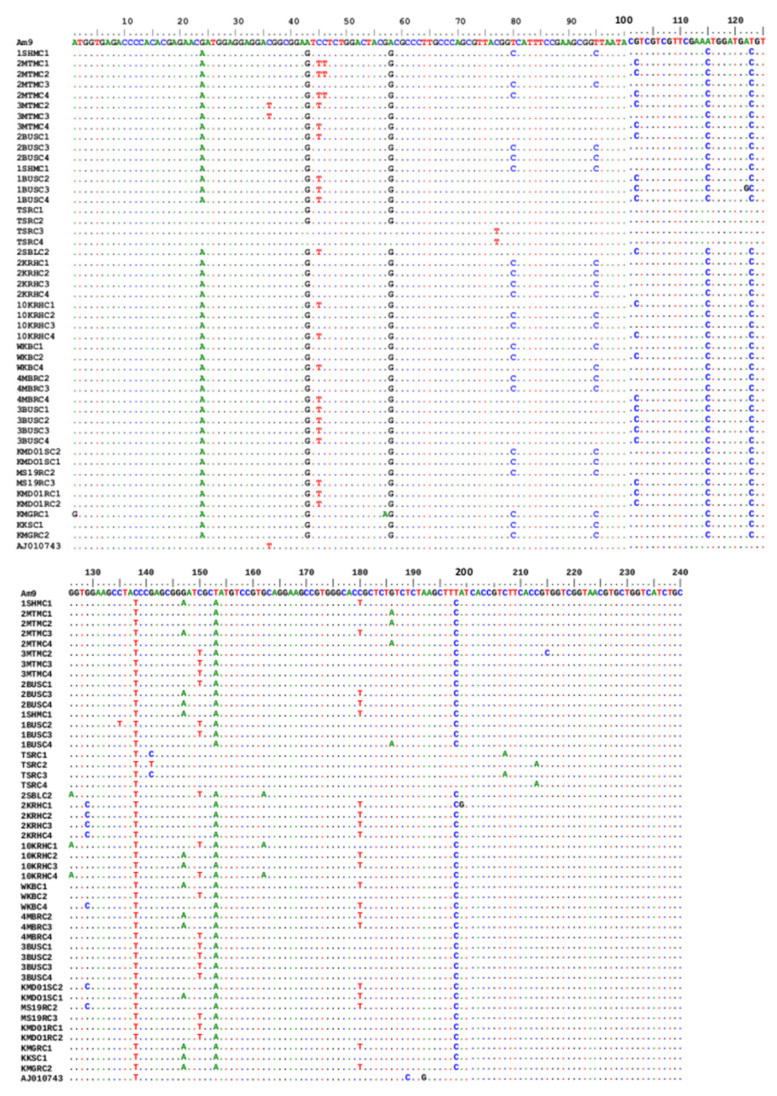
Multiple sequence analysis of 231 bp *R.* (*B*.) *decoloratus* OCT/Tyr receptor gene partial coding region in amitraz-resistant and -susceptible ticks. AJ010743, octopamine-like, G-protein coupled receptor from *R.* (*B*.) *microplus* strain from Australia. A total of 47 sequences were generated in this study. Amitraz resistance was associated with SNPs at loci A1G, C46T A122G and T215C. AM9 to KMGRC2 identifies different clones of the OCT/Tyr gene from various tick populations studied.

**Figure 4 microorganisms-10-02384-f004:**
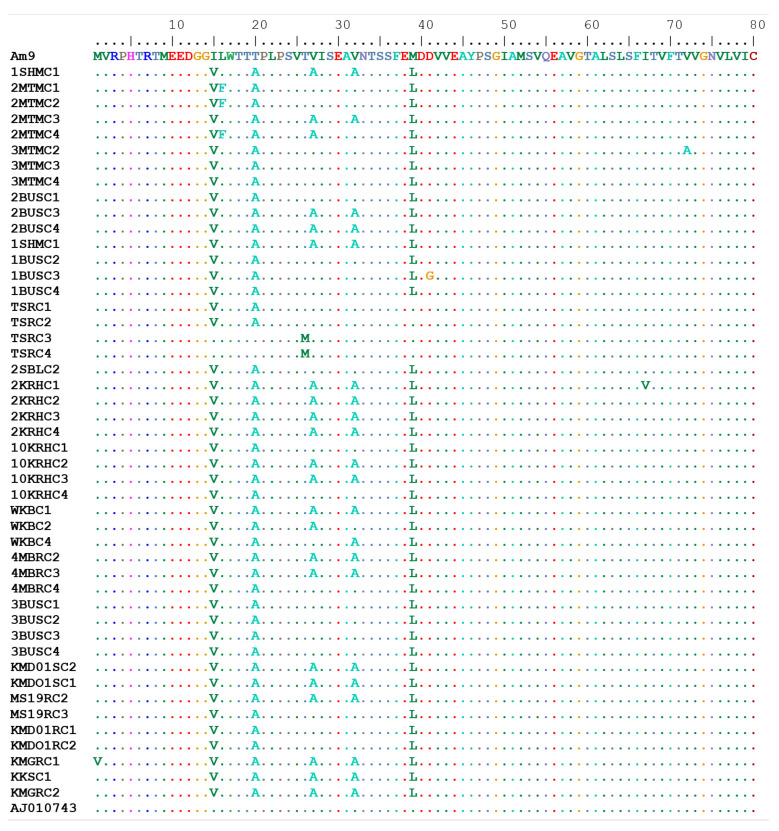
Multiple sequence alignment of deduced amino acid sequence for *R.* (*B*.) *decoloratus* OCT/Tyr receptor gene partial coding region in amitraz-resistant and -susceptible ticks. OCT/Tyr sequence from GenBank: AJ010743.1 (octopamine-like, G-protein coupled receptor from *R.* (*B*.) *microplus* strain from Australia). Amitraz resistance was associated with M1V, F16F, D41G and V72A amino acid substitutions. AM9 to KMGRC2 identifies different clones of the OCT/Tyr gene from various tick populations studied.

**Figure 5 microorganisms-10-02384-f005:**
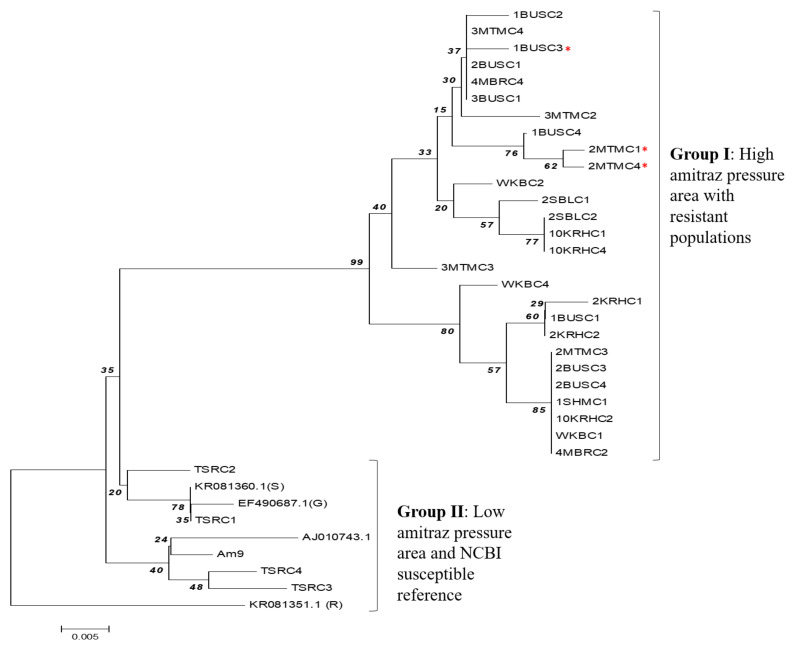
Phylogenetic tree constructed from *R.* (*B*.) *decoloratus* OCT/Tyr receptor gene partial coding nucleotide sequences. Group I, OCT/Tyr gene sequence from tick populations from high acaricide pressure area (central and western Uganda); Red asterisk (*) indicates clones from tick populations with history of amitraz resistance. Group II, tick populations from low acaricide pressure/control area (east and northern Uganda); OCT/Tyr sequence from GenBank: KR081360.1 (S), Susceptible *R.* (*B*.) *microplus* from South Africa; EF490687.1(G), Susceptible *R.* (*B*.) *microplus* Gonzalez strain (America); KR081351.1 (R), amitraz-resistant *R.* (*B*.) *microplus* from South Africa. AJ010743.1 (octopamine-like, G-protein coupled receptor from *R.* (*B*.) *microplus* strain from Australia). The maximum composite likelihood method was used to construct the phylogenetic tree with bootstrap value of 1000 replicate. The tree was constructed using 36 nucleotide sequence, 32 of which were generated in this study.

**Table 1 microorganisms-10-02384-t001:** Characteristics of the farms from which *Rhipicephalus* (*B*.) *decoloratus* ticks were collected.

Region (Number of Districts)	District (Number of Farms)	Farm ID/Tick Population	Sub County	Acaricide Application Interval/Week	History of Acaricides Used in the Last 2 Years	Resistance to Amitraz Based on LPT Assay	Number of Oct/Tyr Clones Sequenced for Each Tick Population
Western (13)	Bushenyi (03)	1BUS	Kyeizooba	Twice	AM + OP + SP	Yes (68.1%)	03
		2BUS	Kyeizooba	-	AM + SP	No	03
		3BUS	Kyeizooba	Twice	AM + SP + COF	No	04
	Kiruhura (05)	2 KRH	Kikatsi	Once	AM	No	04
		10 KRH	Kitkatsi	Once	AM + SP + OP + COF	No	04
		KKS	Kazo	Twice	AM + COF	Yes (0%)	01
		KMD01	Keishunga	Once	AM + COF	Yes (8.0%)	04
		KMG	Keishunga	Twice	AM + COF + OP	Yes (26.7%)	02
	Gomba (01)	MS19R	Madu	Once	AM +SP	Yes (14.5%)	02
	Mbarara (01)	4 MBR	Ndeija	Once	SP	No	03
	Mitoma (02)	2MTM	Mitooma	Once	AM	Yes (45.3%)	04
		3MTM	Kashenshero	Once	OP + SP + COF	Yes (45%)	03
	Sheema (01)	1SHM	Kagango	Once	AM + COF	No	01
Central (02)	Sembabule (01)	2 SBL	Lugusulu	-	AM + OP + COF	No	01
	Wakiso (01)	WKB	Wakiso	Twice	AM + COF	No	03
East (01)	Serere (01)	TSR ^Þ^	Serere	Once	SP	No	04
**North (01)**	**Adjumani (01)**	**AM9 ^Þ^**	**Pachara**	**Occasional**	**AM**	**No**	**01**
**Total**	**Farms (17)**						**47**

LPT, larval packet test; Both (Local and improved cattle). ^Þ^ No amitraz used in the last 2 years; ^Þ^ Control populations from low acaricide pressure area of Uganda, (-) not disclosed; Tick populations used in this study were archived samples and amitraz susceptibility was determined in our previous study [27,31]. AM, amidine (amitraz); OP, organophosphate (chlorfenvinphos); SP, synthetic pyrethroid (cypermethrin or deltamethrin); COF, co-formulation containing organophosphate (chlorfenvinphos or chlorpyriphos) and synthetic pyrethroid (cypermethrin).

**Table 2 microorganisms-10-02384-t002:** Non-synonymous mutations in the partial coding region of *Rhipicephalus* (*B*.) *decoloratus* OCT/Tyr gene and corresponding substitution in amino acid residues.

Category	SNP Loci in ORF	SNP in ORF	SNP Loci Based on Previous Nomenclature by Chen et al. [26] and Baron et al. [12]	Amino Acid Substitution	Phenotype Status
Low amitraz pressure area	Locus 43	A43G	A178G	I15V	Susceptible
	Locus 58	A58G	A193G	T20A	Susceptible
	Locus 77	C77T	C212T	T26M	Susceptible
High amitraz pressure area	**Locus 1**	**A1G**	**A135G**	**M1V**	**Resistant**
	**Locus 46**	**C46T**	**C181T**	**L16F**	**Resistant**
	Locus 80	T80C	T215C	V27A	Susceptible
	Locus 95	T95C	T230C	V32A	Susceptible
	Locus 115	A115C	A115C	M39L	Susceptible
	**Locus 122**	**A122G**	**A257G**	**D41G**	**Resistant**
	Locus 199	A199G	A334G	I67V	Susceptible
	**Locus 215**	**T215C**	**T350C**	**V72A**	**Resistant**

ORF: Open reading frame; Bold font: Nucleotide and amino acid substitutions found in amitraz-resistant tick populations. Low amitraz pressure area farm ID: AM9, TSR. High amitraz pressure area farm ID: 1BUS, 2BUS, 3BUS, 2KRH, 10KRH, 4MBR, 2MTM, 3MTM, 1SHM, 2SBL, WKB.

## Data Availability

The nucleotide sequences generated in this study have been deposited in GenBank for public access.

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
