# Peer review of "Molecular Characterization of Octopamine/Tyramine Receptor Gene of Amitraz-Resistant Rhipicephalus (Boophilus) decoloratus Ticks from Uganda"

_microorganisms, 2022, doi:10.3390/microorganisms10122384_

Round 1

Reviewer 1 Report

The manuscript "Molecular Characterization of Octopamine-Tyramine receptor gene of Amitraz Resistant Rhipicephalus (Boophilus) decoloratus Ticks from Uganda" describes the characterization of receptor gene and reports four novel amino acid mutations that are associated with phenotypic amitraz resistance in Uganda. This study could help with pesticide control in Uganda and other countries where ticks represent a significant source of harm to livestock farmers. The considerations are presented annexed.

Author Response

RESPONSE TO COMMENTS BY REVIEWER 1

Dear Editor,

The authors of the manuscript ID-Microorganisms-2006140 would like to thank the reviewers for the feedback and suggestions aimed at improving the paper. Additional edits made to the manuscript is shown in track changes (SEE ATTACHED FILE). Since Reviewer 2 had no concerned raised, here below we provided point-by-point response to reviewer 1.

Reviewer

 For example, in lines 31-33: In conclusion, this study is the first to characterize R. (B.) decoloratus OCT/Tyr receptor gene and reports four novel amino acid mutations that are associated with phenotypic amitraz resistance in Uganda.

Better: In conclusion, this study is the first to characterize the R. (B.) decoloratus OCT/Tyr receptor gene and reports four novel amino acid mutations associated with phenotypic amitraz resistance in Uganda.

Response

The suggestion was implemented in the revised manuscript

Reviewer

 The sample used is small. The description could be more detailed.

Response

The authors acknowledge the few samples that may appear as a limitation to the study, the details of the location at both district and sub county level where the ticks were collected is shown in Table 1.

Reviewer

 How were these larvae populations obtained?

Response

As indicated in the manuscript, the larvae were obtained by incubation of fully engorged female ticks in the laboratory, where they laid eggs and hatched to larvae.

Reviewer

How many female ticks were used to obtain the larvae?

On average, 5-10 fully engorged female ticks were incubated to obtain the larvae

Reviewer

The regions where resistant and susceptible larvae were collected could be marked in different colors on the map.

Response

As advised, different colors were used to depict the districts from which samples were collected; with RED representing districts from high acaricide pressure area while purple for districts from low acaricide pressure area

Reviewer

The genetic composition of cattle breeds from herds with resistant and susceptible parasites should be better described in terms of the degree of Taurine and Zebu blood.

Response

The authors appreciated the suggestion, however, it was not possible to obtain accurate degree of breed composition to the level of degree of Taurine & blood since it was beyond the scope of the study.

Reviewer

 It is necessary to describe which other insecticides were used either before or concomitantly.

Response

We have included additional column in Table 1 to capture the classes of acaricides used on the farm in the last 2 years in the revised manuscript.

Reviewer

The numbers of sequenced clones were tiny.

Response

The authors appreciate the concern raised by the reviewer, but at the same time recognize this as potential weakness of the study.

Extraction of genomic DNA:

Reviewer

Including a negative control and describing degrees of purity and concentrations would be necessary

Response

The authors reiterate that R. microplus ticks was used as control and all the resultant DNA eluted during extraction had purity ratio of above 1.8. A statement was included in the revised manuscript on the purity ratio of the DNA extracted as guided by the reviewer.

Amplification of octopamine/tyramine receptor gene:

Reviewer

It would be necessary to include positive and negative control DNA.

Response

The R. microplus genomic DNA was used as a positive as shown in Fig 2. For every run, de-ionised water was used as a negative control and no amplicon resulted from it although not included in the gel image.

Results

Reviewer

Figure 3 – needs to improve the definition.

Response

Figure 3 has high resolution except that due to compression to accommodate the length size of the file, it appears to have low resolution.

Discussion:

Reviewer

Line 242: Amitraz is considered a cornerstone in controlling resistant ticks. It would be interesting to provide more information about this claim.

Response

The advice was taken and the statement revised to explain the basis of amitraz being a cornerstone in the manuscript.

Reviewer 2 Report

Reviewer report:

The manuscript Patrick Vudriko, Rika Umemiya-Shirafuji, Dickson Stuart Tayebwa, Joseph Byaruhanga, Benedicto Byamukama, Maria Tumwebaze, Xuenan Xuan and Hiroshi Suzuki.

The authors described the Molecular Characterization of Octopamine-Tyramine receptor 2 gene of Amitraz Resistant Rhipicephalus (Boophilus) decoloratus Ticks from Uganda. Overall, the study is well described in both methodology and results sections, and the information is interesting and important for researchers in the field of acaricide resistance in ticks and also tick-borne diseases.

No revisions are suggested to improve the manuscript. The manuscript should be submitted in its current form.

Author Response

The authors would like to thank the reviewer for the positive feedback